# Infrared Thermography for Early Identification and Treatment of Shoulder Sores to Improve Sow and Piglet Welfare

**DOI:** 10.3390/ani12223136

**Published:** 2022-11-14

**Authors:** Lauren M. Staveley, Jessica E. Zemitis, Kate J. Plush, Darryl N. D’Souza

**Affiliations:** SunPork Group, 1/6 Eagleview Place, Eagle Farm, Brisbane, QLD 4009, Australia

**Keywords:** shoulder sores, lesion, ulcer, sow welfare, infrared thermography (IRT)

## Abstract

**Simple Summary:**

Shoulder sores are a welfare issue primarily for breeding sows and are often associated with reduced movement and body condition loss during lactation. The occurrence of shoulder sores contributes to lactation failure, which in turn becomes a welfare issue for piglets due to the stress of early weaning. This experiment investigated the use of a thermal camera to detect hotspots caused by underlying tissue damage on the shoulder of sows, prior to wound formation. Infrared thermography (IRT) was able to detect hotspots in 88% of sows that developed sores, allowing for a lead in time to wound formation of 7 days. Once a hotspot was detected, we investigated the use of three commonly used topical commercial products, Repiderma^®^, Derisal^®^, Chloromide^®^ and mānuka honey. All treatments, with the exception of Repiderma^®^, decreased the diameter of the sores by weaning. The results of this study provide evidence that the use of IRT provides a tool for early detection of shoulder sores in sows, allowing for early treatment that is likely to improve the longevity of the sow and welfare of the litter, due to a reduction in lesion related sow culling.

**Abstract:**

Shoulder sores in sows cause pain, may lead to early weaning and resultant piglet distress, and premature culling. Early detection and intervention is key to avoid these substantial production and welfare imposts. In this experiment we tested infrared thermography (IRT) to identify shoulder sores before wound eruption, and effectiveness of four wound healing treatments; mānuka honey (*n* = 11); Derisal^®^ (*n* = 11); Chloromide^®^ (*n* = 13) and Repiderma^®^ (*n* = 16), all of which contain no antibiotics. Three hundred and ten sows (parity 2.3 ± 0.2) were monitored daily from entry to the farrowing sheds until weaning using a thermal camera. IRT successfully detected 88% of shoulder sores as hot spots and provided a lead in time of 7 days. Sixteen percent of all sows had a hotspot detected and were randomly allocated to one of four daily treatment groups. At the end of the treatment period, sore diameter was significantly reduced for all treatments, except the Repiderma^®^ group. Sow traits had little influence on susceptibility to shoulder sores. There may be a link between prewean mortality and shoulder sores, but this requires further investigation. The use of IRT to monitor for hotspots for early intervention is validated. Future work should concentrate on methods to prevent wound eruption after detection with IRT to improve the health and welfare of both the sow and her litter.

## 1. Introduction

Shoulder sores develop primarily in breeding sows and frequently lead to premature culling, with prevalence reported as highly variable, ranging from 5 and 50% [1]. The prevalence of shoulder sores increases when sows are housed in farrowing crates as recumbency on a solid surface for extended periods of time will cause damage to the skin, allowing development of sores. Sores are more likely to develop in sows with poor body condition, or in those individuals’ experiencing disease or injury such as lameness [2]. In addition, scar tissue on the shoulder caused by a previous sore, greatly increases the probability of future sores developing [3,4]. Naturally ventilated sheds rely on evaporative cooling, which involves a water dripper above each crate activated when ambient temperatures are high. This is a widely used system to cool sows in the warmer months in Australia, however the moisture this creates between the floor and the shoulder of the sow often exacerbates the incidence of shoulder sores. While precautions are taken, such as plastic slatted flooring to facilitate airflow and drying, it still presents an issue. In addition to the increased moisture, feed intake is also reduced in the warmer months which increases the incidence of shoulder sores in two ways; it increases the time spent in recumbency and leads to greater weight loss in lactation.

Shoulder sores are detected visually by lesions present on the surface of the skin. However, underlying tissue damage usually begins several weeks prior to formation of a visible sore [5]. Once identified, sores are generally treated using antibiotic administration and topical ointment to encourage wound healing shortly after weaning [6]. In severe cases, sows are removed early from lactation to halt tissue damage, ensuring that the sow’s welfare is not compromised [5]. When sows are removed from farrowing accommodation early there is a requirement for foster sow usage, or the early weaning of piglets which can cause distress for the piglets. The treatments/interventions for shoulder sores are labour intensive, time-consuming, expensive, and contribute to higher antibiotic usage. Identifying shoulder sores in the early stages, when tissue degradation is beginning, would be beneficial as it would facilitate early management, treatment and healing. Additionally, this would reduce labour requirements for treatment application and decrease the culling rates associated with poor body condition or lameness.

New technologies for the early detection of compromised animal health and welfare under research conditions are well reported, but application under commercial conditions is less frequent. Infrared thermography (IRT) can be used to detect elevated blood flow and the associated temperature rise due to an inflammatory response. This technology has been effective in detecting increased blood flow and the associated temperature rise in foot lesions of sows, and successfully used for early detection of lameness [7]. However, in sows the individual variation in mean shoulder temperature is too great to allow for easy identification of at-risk sows, therefore, the use of hot spot (i.e., a local temperature increase) detection is required [5]. While these results are promising, they have not yet been tested under commercial conditions. 

Topical products that effectively target sores and encourages tissue repair would be beneficial when coupled with early detection of shoulder sores using thermal imaging to achieve rapid healing. Currently available products include Chloromide^®^ antiseptic spray, which is an insect repellent used to treat infections and bacterial skin diseases. Derisal^®^, designed as a topical ointment to provide a barrier whilst treating general skin infections and Repiderma^®^, an antimicrobial protective aerosol. Mānuka honey is increasingly recognised in both human and veterinary medicine as an effective alternative for the treatment of conditions including limb ulcers, abscesses and otherwise difficult to manage wounds [8]. It is inhibitive to a range of bacterial pathogens including those that can colonise the skin, and there is no known evidence of microbial resistance to honey. It’s antimicrobial and anti-inflammatory properties position it well as an alternative for shoulder sore treatment [9].

The aim of this study was to determine if the use of IRT to detect tissue degradation prior to visible surface signs is an effective early detection tool in a commercial setting. Detected cases provided the cohort to determine which course of topical treatment was most effective in resolving shoulder sores. 

## 2. Materials and Methods

### 2.1. Farm

This study was conducted on a commercial breeder unit, in South Australia which largely experiences a Mediterranean climate. The breeder unit houses 7500 sows and suckling piglets. The farm operates on a continuous farrowing basis. The genetic make-up of the farm was largely F1 crossbred sows (*n* = 237) however the herd is self-replacing and as such purebred sows (*n* = 43) are present. This experiment ulitised both crossbred and purebred sows. 

### 2.2. Animals and Housing

This study was conducted on a breeder unit with experimental sows (PIC Camborough 42) mated in July and August and farrowed in November (THI = 68) and December (THI = 65). Sows were moved into two farrowing sheds five days prior to expected farrowing date. These sheds were identical in design with treatments equally replicated, and all sows farrowed with a 7 day period. The farrowing house was naturally ventilated, with a temperature (28 °C) activated dripper system for evaporative cooling. Housing consisted of a standard farrowing crate 1.8 × 2.4 metres in size with a creep area heated via lamp for piglets, an *ad libitum* feeder, two water nipples for the sow and one for the piglets. Flooring of crates consisted entirely of plastic slatted tiles. Sows were delivered 2 kg of feed twice daily, at 0700 and 1600 prior to farrowing, with *ad libitum* access to a standard lactation diet post farrowing. At 24 h of age, piglets were cross fostered based on the sows rearing capacity (functional teat number) and on day 2 were tail-docked, administered an iron injection and an oral coccidiostat. Sows and piglets were weaned at 23.0 ± 0.2 days of age.

### 2.3. Measurements

Three hundred and ten sows (parity 2.3 ± 0.2) were observed from entry to the farrowing house until weaning. Each sow received a body condition score on a scale of 1 to 5, where 1 was severely underweight and 5 severely overweight (Model Code of Practice for the Welfare of Animals; Pigs 3rd Edition) upon entry to the farrowing house to ensure balanced allocation to treatment group. Ultrasonic backfat measured at P2 site on the last rib on the right side (Imago.S, IMV imagine, Rochester, NY, USA) was measured upon entry and at exit to the farrowing house. A thermal image of both shoulders (left and right) from each sow was taken upon entry to the farrowing house and every three days until weaning using a FLIR thermal imaging camera (FLIR T300 camera, 2008; FLIR Systems, Boston, MA, USA). Thermograph resolution was calibrated to ambient temperature before each image collection and images were scanned with a 0° angle to the shoulder. The images were taken using a 20 × 20 cm section at a distance of one metre from the skin surface, to maintain consistency in image size. The mean, minimum and maximum temperature of the shoulder images were recorded, in addition to the presence of any temperature hot spots. A temperature hot spot was defined as the sow showing a local temperature increase over the shoulder [5] (Figure 1). If a temperature hotspot was detected, thermal images were recorded daily to assess healing.

### 2.4. Treatments

Sows presenting with a shoulder sore were allocated to one of the four treatment groups, ensuring even BCS and parity distribution. The four treatment groups included topical Derisal^®^ (Zinc Oxide 30 g/kg, Boric Acid 15 g/kg. Rudduck Australia Pty Ltd., Keysborough, VIC, Australia; *n* = 11), 100% mānuka honey (UMF 20+, The Better Health Company Pty Ltd., Mount Waverley, VIC, Australia; *n* = 11), Repiderma^®^ topical spray (Copper amino acid chelate 5%, Zinc amino acid chelate 6.8%, E-100 [extract] 0.02%. Intracare BV, Veghel, The Netherlands; *n* = 16), or Chloride medicated spray (Chloromide^®^, Centrimide 8.0 g/L, N-Octyl bicycloheptene dicarboximide 5.0 g/L, Di-N- propylisocinchomeronate 2.5 g/L, Chloroxylenol 2.4 g/L, Orthophenylphenol 0.8 g/L, Pyrethrins 0.3 g/L. Troy laboratories; Glendenning, NSW, Australia; *n* = 13). Treatments were applied daily, as per specifications on the label, until the wound repaired, with sore diameter recorded prior to each application using Vernier calipers (Absolute Digimatic Caliper, Mitutoyo, Kawasaki, Japan). All shoulder sores were rated using a scaling system of 1 to 4 based on severity at the same time (Table 1).

All production measures were recorded, including total piglets born, born alive, born dead, and number of pigs weaned.

### 2.5. Statistics

Data was analysed in SPSS v25 (IBM, Armonk, NY, USA) and a *p*-value < 0.05 considered significant. Predictors/impacts of shoulder sores were analysed using negative binomial regression (count data) and general linear model (continuous data) with sow location (shed 1 or 2) and shoulder sore (no or yes) as fixed effects. The effects of shoulder sore presence on production measures were analysed according to the following model:Y = β_0_ + β_s_ + ε
where Y denotes the production measures, β_0_ is the fixed effect of location, β_s_ is the fixed effect of the presence or absence of a shoulder sore, and ε is the normally distributed residual error.

The impact of treatments was analysed using negative binomial regression (count data) and general linear model (continuous data) with sow location (shed 1 or 2) and treatment (Derisal^®^, mānuka, Repiderma^®^ or Chloromide^®^) as fixed effects. As the start diameter of a shoulder sore was significantly different between treatments, this measure was fitted as a covariate for end diameter, change in diameter, and largest diameter. Severity scores (start, end, change and highest) did not meet assumptions of normality and so were analysed using Kruskal–Wallis test. Proportion of sows with healed shoulder sores by weaning was analysed using chi-squared test. The effects of treatment on shoulder sores were analysed according to the following model:Y = β_0_ + β_s_ + X_cdrm_ + ε
where Y denotes the response to treatment, β_0_ is the fixed effect of location, β_s_ is the fixed effect of the presence or absence of a shoulder sore, X is the treatment (where c = Chloromide^®^, d = Derisal^®^, r = Repiderma^®^ and m = mānuka), and ε is the normally distributed residual error.

## 3. Results

### 3.1. Early Detection of Shoulder Sores

The fixed effects of shed were not significant for any traits analysed. Of the 310 sows included in this study, none presented with a visual shoulder sore prior to farrowing, but there was a 16% incidence in shoulder sores over lactation, with 51% of those recovering with treatment prior to weaning. The use of IRT detected hotspots on the shoulders of 88% of sows who later presented with sores, and this hotspot was observed 6.96 days prior to sore formation (Table 2).

Parity, number of piglets born or P2 back fat upon entry (Table 3) were not predictors of shoulder sore incidence. The time a sow spent in a crate prior to farrowing did affect the occurrence of shoulder sores, with sows who developed a sore being in a crate for 0.5 days less before farrowing (*p* < 0.05). The number of piglets reared by a sow, lactation length, exit or change in P2 backfat (Table 3) did not impact shoulder sore formation.

### 3.2. Treatment of Shoulder Sores

The start diameter of a sore was significantly lower in the group of sows treated with Chloromide^®^ (Figure 2), as such start diameter was fitted as a covariate for analysis of all other diameter measurements. Derisal^®^, mānuka and Chloromide^®^ treatment all significantly reduced the diameter of the shoulder sore (*p* < 0.05), while Repiderma^®^ had no effect. 

Severity score of lesions was analysed using the Kruskal–Wallis H test, with results presented as such. There was no effect of treatment on the start severity score of shoulder sores (H(3) = 0.687, *p* = 0.876), the end severity score (H(3) = 2.419, *p* = 0.490), change in severity (H(3) = 1.626, *p* = 0.654) or the highest severity score (H(3) = 3.462, *p* = 0.326). The percentage of healed cases in each treatment group was not significantly different (Derisal^®^ 45.5%; mānuka 63.6%; Repiderma^®^ 37.5%; Chloromide^®^ 61.5%, chi squared X(3) = 0.258, *p* = 0.461).

## 4. Discussion

Incidence of shoulder sores in the current investigation was 16% and so is an issue that impacts a significant number of breeding sows around farrowing. IRT can be used to detect shoulder sores 88% of the time, and were detected 7 days prior to sore eruption, meaning that early intervention strategies can be applied. After wound eruption, successful treatments tested were daily topical application of mānuka honey, Chloromide^®^ or Derisal^®^. Findings from this investigation can be applied as an early warning tool for the prevention of lactation related incidences of shoulder sores, improving sow and piglet welfare.

Shoulder sore prevalence is very variable and ranges from 5 and 50% [1]. The current investigation was conducted in November and December, with an average daily maximum temperature of 29.9 ± 0.8 °C. These types of high temperatures commonly observed during the summer months in Australia, are often associated with decreased postural changes of sows, increasing the risk of shoulder sores. This increased time spent in lateral recumbency may also explain the trend for a reduction in piglet deaths in sows with shoulder sores, as fewer posture changes decreases the risk of piglets being overlain. 

Prevention should be the primary aim of shoulder sore management. This can involve numerous methods with the most important being the maintenance of body condition leading up to and during lactation. This can be achieved through close monitoring of feed quantity, consumption, and nutritional quality, as sows that lose significant body condition during lactation tend to be the sows with the most severe sores [2]. There was no effect of condition loss on incidence of shoulder sores in this study, as sows only lost an average of 0.6 ± 0.1 mm of P2 backfat during lactation. Surprisingly, this study demonstrated that increased time (~0.5 days) in a crate prior to farrowing reduced the likelihood of a sow developing shoulder sores. Sows were housed on concrete flooring in gestation and perhaps the increased time allowed the sow to better adapt to the plastic slatted flooring prior to farrowing. Previous research has shown that drastic changes in housing and flooring types between gestation and farrowing increases the risk of lameness in sows [10], which may exacerbate the time spent in lateral recumbency. 

The results of this study provide evidence that the use of IRT allows early intervention, with the ability to detect 88% of sores from one meter away, 7 days before wound eruption. This form of detection is relatively simple, requiring minimal training of staff and allows for early detection and timely intervention such as those investigated. The implementation of such technologies into the daily monitoring of sows, particularly in the farrowing house, would create a novel and proactive approach to the care and treatment of shoulder sores. The regular inspection of sow shoulders with IRT would require additional labour but could easily be embedded in routine animal monitoring. While the thermal cameras used in this study were costly and large, there has since been the development of a small phone or tablet plug-in camera that connects to any smart device, allowing for a more cost-effective option. The use of IRT is a simple and easily implementable tool that can have potentially vast impacts on the welfare of sows, particularly during the summer months when the prevalence of shoulder sores is exacerbated.

This study investigated the use of three commonly used topical commercial products, Repiderma^®^, Derisal^®^ and Chloromide^®^, in addition to mānuka honey, previously used in alternative human medicine [8]. At the end of the treatment period, sore diameter was significantly reduced for all treatments, except the Repiderma^®^ group. While mānuka honey provided similar healing results to Derisal^®^ and Chloromide^®^, its commercial applicability is questionable, due to the cost and ease of use explored below. Significance was not established on the healing rate of each product tested, likely due to the low sample size for a binary trait. The numerical differences between treatment warrant further investigation that follows the sows through to gestation and with a larger sample size. There was no significant difference in end diameter of the sores between the Derisal^®^, mānuka and Chloromide^®^ treated groups, therefore the best commercial recommendation would be to use Chloromide^®^ in the future, as it showed the greatest potential for rapid healing, is affordable and easy to apply.

The effects of mānuka honey on second intention healing of lower limb wounds in horses have shown that wounds retracted less and healed faster than untreated wounds [11]. Results from this study agree with these previous findings [11], as daily application of mānuka honey to shoulder sores in sows reduced the diameter of these sores, with no antibiotic intervention, and resulted in 63.6% of sows with healed wounds by weaning. Interestingly, it was observed that interest from piglets and flies coupled with the water-cooling system used on the experimental site resulted in a low contact time between the wound and the honey. Despite these issues resulting in a lowered contact time for mānuka compared to the other treatments, it was still an effective treatment for shoulder sores. Natural replacements for antibiotics are becoming more commonplace in agricultural industries, and while mānuka is not a practical option in its current state, there is potential for refinement of the product, by reducing its attractiveness to insects and other pigs.

Treatment with Derisal^®^ and Chloromide^®^ achieved similar rates of healing as mānuka honey, and the efficacy of these treatments may be improved, as they have been designed to repel insects and reduce interference by other animals, due to their bitter taste. While the use of Derisal^®^ as a treatment for shoulder sores is effective, as it is a cream, it does not provide ease of use comparable to Chloromide^®^. Previous work by Hallet et al. [12] found that an inclusion rate of 15.25% zinc oxide to topical treatments provided significantly improved healing rates to Chloromide^®^ which contrasts the current findings. As Derisal^®^ consists of only 3% zinc oxide, it is possible that higher inclusion rates are required to improve sore healing than those reported here. Repiderma^®^ spray was found to be ineffective in the treatment of shoulder sores, as the sore diameter did not change, while all other treatments decreased. Anecdotally, sows from this treatment were observed to rub the wound on crate fixtures after application, which may result in further damage to the sore and a reduction in contact time between the active ingredients and wound, but also indicates this product resulted in pain after administration [13]. 

The results of this study provide evidence that the use of IRT allows for detection of shoulder sores prior to sore eruption and as an early detection method with a significant lead in time. Future work should focus on prevention by reducing the time a sow spends in lateral recumbency once a hot spot is detected. The use of mānuka based therapies should be considered for treatment of hotspots, as current silk [14] and pectin [15] membranes combined with mānuka are hydrophobic and proving to be very effective antibacterial agents, preventing >98% of bacterial growth. 

## 5. Conclusions

The outcomes of this experiment have demonstrated that the use of IRT can effectively identify sows at greater risk of developing shoulder sores prior to wound eruption. A significant number of breeding sows are impacted by shoulder sores as the incidence in the current investigation was 16%. Therefore, the implementation of IRT should be adopted industry wide as it allows for 7 days lead in to wound eruption in 88% of cases. After wound eruption, the most successful treatment tested was daily topical application of mānuka honey, Chloromide^®^ or Derisal^®^. Findings from this investigation can be applied as an early warning tool to monitor for emergence of this condition within a herd, improving lactating sow and piglet welfare.

## Figures and Tables

**Figure 1 animals-12-03136-f001:**
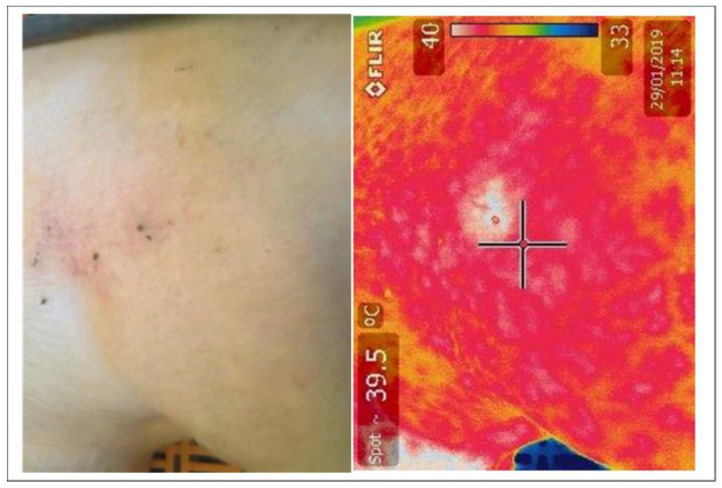
Thermal image of hotspots identified on the shoulder of a sow using FLIR thermal imaging camera.

**Figure 2 animals-12-03136-f002:**
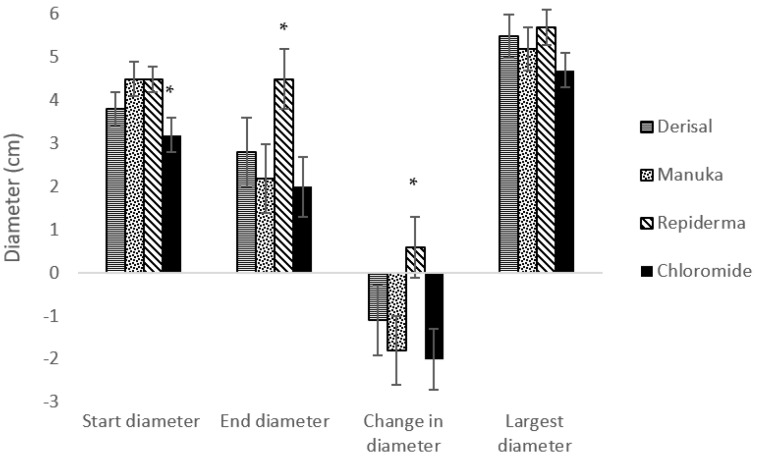
Mean ± SEM of sore diameter characteristics for sows treated daily with topical treatments of Derisal^®^, mānuka, Repiderma^®^ or Chloromide^®^. Asterix (*) signifies a significance of *p* ≤ 0.05. Start and end diameter refers to the diameter of the shoulder sore at the start and end of the treatment period, with the change being the end minus the start diameter.

**Table 1 animals-12-03136-t001:** Grading system (severity score) used for assessing sores on clinical inspection [5].

	Score	Definition
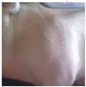	0	No sore.
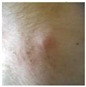	1	Small wound, limited to the outer layer of the skin (epidermis)
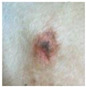	2	Broken skin surface and destruction of the lower layer (dermis)
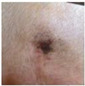	3	Large sores with underlying skin affected, and granulated tissue formed
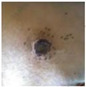	4	Sores extending into the underlying bone

**Table 2 animals-12-03136-t002:** Mean ± SEM values for timing, occurrence, detection, and healing of shoulder sores.

	*n*	Mean	SEM
Incidence (proportion)	310	0.16	0.12–0.21 *
Age at detection (lactation days)	51	8.78	0.74
Number of treatments	51	10.08	0.67
Healed by weaning (proportion)	51	0.51	0.79–0.97 *
Detected by IRT (proportion)	51	0.88	0.37–0.65 *
Detected before visible sore (days)	51	6.96	0.66

* 95% CI presented rather than SEM presented for binary data.

**Table 3 animals-12-03136-t003:** Mean ± SEM of production data, piglet mortality, time spent in the farrowing accommodation and P2 backfat changes. Significance considered *p* ≤ 0.05.

	No Shoulder Sore		Shoulder Sore		*p* Value
	Mean	SEM	Mean	SEM	
*n*	259		51		
Parity	2.5	0.1	2.4	0.1	0.771
Total pigs born (*n*)	13.7	0.9	13.5	2	0.981
Pigs born alive (*n*)	12.8	0.8	12.8	1.9	0.999
Pigs born dead (*n*)	0.9	0.1	0.7	0.2	0.287
Mummified pigs (*n*)	0.2	0.1	0.3	0.1	0.657
Pigs after fostering (*n*)	12.1	0.8	12.1	1.8	0.994
Number of pigs weaned (*n*)	10.5	0.7	10.9	1.6	0.808
Deaths before fostering (*n*)	0.5	0.1	0.3	0.1	0.292
Deaths after fostering (*n*)	0.9	0.1	0.6	0.1	0.137
Liveborn deaths (*n*)	1.3	0.1	0.9	0.2	0.133
Total deaths (*n*)	2.2	0.2	1.6	0.3	0.112
Piglet removal for ill thrift (*n*)	0.7	0.1	0.5	0.1	0.447
Time in crate before farrowing (days)	6.9	0.1	6.4	0.2	0.037
Lactation length (days)	22.9	0.1	23	0.3	0.962
Entry P2 (mm)	19.1	0.1	19.4	0.3	0.276
Exit P2 (mm)	18.5	0.1	18.6	0.3	0.711
P2 change (mm)	0.6	0.1	0.9	0.3	0.405

## Data Availability

The data presented in this study are available on request from the corresponding author.

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
