# Peer review of "Infrared Thermography for Early Identification and Treatment of Shoulder Sores to Improve Sow and Piglet Welfare"

_animals, 2022, doi:10.3390/ani12223136_

Round 1

Reviewer 1 Report

Shoulder sores of breeding sows cause damage and can affect the health and welfare of piglet as well. Early detection is recommended for an early application of treatments and rapid healing of wounds avoiding greate evils.

From a few days before farrowing and during lactation, hot spots prior to the formation of wounds are located using infrared thermography, allowing early treatment of sores, which improves the welfare of the sow and the litter.

This work studies how infrared thermography detects shoulder sores and how it affects to the wounds, on the other hand various treatments are contrasted to combat sores on the shoulders

MATTERS TO CLARIFY

LINE 88 Detail how many crossbred and how many purebred sows are used in the test.

Line 97 2 kg. In each intake or in total?

LINE 104 Body condition was measured. Does not appear in results. What was it measured for? To assign the sows to each treatment? indicate it.

LINE 150 Why do the fixed effects (Shed 1, Shed 2) not appear in the results?

LINE 153 Explain in more detail: end diameter, change in diameter and largest diameter

Table 2 Detected before visible sore. It is understood that in all the hot spots detected, sores emerged. So there was no treatment to prevent the appearance of sores.  It is right?

LINES 194-197 Explain what H(3) is

Author Response

Thank you for taking the time to review and comment upon our manuscript, animals-1932691 “Infrared thermography for early identification and treatment of shoulder sores to improve sow and piglet welfare”. We found the advice constructive and have incorporated your suggestions into our revision.

We have responded to each comment individually in the attached word document.

Reviewer 2 Report

·      Authors need to cut down and prioritize the key words according to the requirement of the study.

·      It would be much better if authors add data about prevalence rate of shoulder sores globally or national further the economic losses due to the same condition. 

·      Authors need to brief about products used products to reduce shoulder sores in introduction. 

·      In technical program section authors haven’t mentioned about history of soreness in the sows.

·      Authors need to mention THI during the study period within the shed and outside shed. To better correlate with feeding behavior. 

·      According to various literatures, BCS have a significant effect on shoulder sore. Authors need to mention regarding the inter-relationship between BCS of the experimental sows and the tendency towards shoulder soar.  

·      According to the authors, what is the justification behind ineffectiveness of Repiderma® spray other than contact period, because even the mānuka honey had less contact period but still it was pretty effective compared to Repiderma® spray.

Author Response

(The authors gave the same response as above.)

Reviewer 3 Report

General comments:

This paper on the use of infrared thermography to predict shoulder sores in sows in advance is interesting, but the overall writing skills of the paper need to be improved. The logic of introduction and discussion is not clear.

1.      The order of reference number is incorrect, please correct it.

Specific comments:

1.      Line 30: Does this article prove it— “the presence of a shoulder sore had no impact on litter performance”?

2.      It is recommended that the proportion of sows with shoulder ulcers in previous references be added to the introduction.

3.      Line 66: what is the “hot spot”? The words need more detailed explanation.

4.      In the introduction only the therapeutic effects of Manuka honey are described, but not the effects of several other medicines, which are recommended to be added.

5.      Line 104-105: Why does the body condition score data for the sow not appear in the results?

6.      Line 125: what was the basis for the assignment of “randomly”, and how to ensure the random allocation?

7.      Line 145-156: The statistics model governing equation should be added.

8.      Line 148: Were the sows in a different barn during the trial? If different sheds, a description needs to be added to the materials and methods and proof that the environmental conditions in the two sheds were similar. Were there any differences in the sows suffering from shoulder sores in the different houses?

9.      The titles of table 2 and table 3 are too long, that should be shorten.

10.   Table 2: Why is there a SEM of incidence of shoulder sores?

11.   Line 204: “After wound eruption, the most successful treatments tested were daily topical application of mānuka honey, Chloromide® or Derisal®.” Why use the “most”?

12.   Line 217: The sow's feed intake is not addressed in the trial results, why does it appear in the discussion?

13.   Line 218-221: the sentence should be moved to introduction.

14.   Line 226-232: Is there any literature to prove that half a day increases the sow's ability to adapt to the floor?

15.   Line 261: what is “these previous findings”, the references should be added.

Author Response

(The authors gave the same response as above.)

Round 2

Reviewer 2 Report

The authors have addressed all my queries satisfactorily